# Rational design of inducible CRISPR guide RNAs for *de novo* assembly of transcriptional programs

Quentin R.V. Ferry[1], Radostina Lyutova[1] & Tudor A. Fulga[1]

CRISPR-based transcription regulators (CRISPR-TRs) have transformed the current synthetic biology landscape by allowing specific activation or repression of any target gene. Here we report a modular and versatile framework enabling rapid implementation of inducible CRISPR-TRs in mammalian cells. This strategy relies on the design of a spacer-blocking hairpin (SBH) structure at the 5′ end of the single guide RNA (sgRNA), which abrogates the function of CRISPR-transcriptional activators. By replacing the SBH loop with ligand-controlled RNA-cleaving units, we demonstrate conditional activation of quiescent sgRNAs programmed to respond to genetically encoded or externally delivered triggers. We use this system to couple multiple synthetic and endogenous target genes with specific inducers, and assemble gene regulatory modules demonstrating parallel and orthogonal transcriptional programs. We anticipate that this 'plug and play' approach will be a valuable addition to the synthetic biology toolkit, facilitating the understanding of natural gene circuits and the design of cell-based therapeutic strategies.

[1] Weatherall Institute of Molecular Medicine, Radcliffe Department of Medicine, University of Oxford, Oxford OX3 9DS, UK. Correspondence and requests for materials should be addressed to T.A.F. (email: tudor.fulga@imm.ox.ac.uk).

Synthetic biology in mammalian systems holds great promise for both deciphering the wiring of natural gene networks (GNs) and engineering cells for therapeutic benefit[1,2]. This process relies on the characterization and assembly of biological parts into *de novo* synthetic pathways designed to redirect or enhance the scope of naturally evolved cellular behaviours[3]. Adding to a growing list of available standardized components, the type-II clustered regularly interspaced short palindromic repeats (CRISPR)/Cas9 from *Streptococcus pyogenes* (*Sp*) has been recently repurposed to create programmable transcriptional regulators (CRISPR-TRs) in mammalian cells[4,5]. CRISPR-TRs rely on the ability to direct a nuclease-deficient Cas9 (dCas9) to any given $N_{20}NGG$ DNA sequence in the genome by simply reprogramming its associated single guide RNA (sgRNA). Consequently, the output expression of any gene of interest can be controlled by tethering various effector domains to the sgRNA–dCas9 complex and targeting them near transcription start sites[4,6] (Fig. 1a).

A critical dimension in synthetic biology is the design of inducible parts, enabling the construction of complex gene circuits responsive to exogenous cues and endogenous metabolites. Although elegant chemically inducible and photoactivated CRISPR/Cas9 solutions were recently reported in mammalian cells, these systems have been restricted to post-translational control of Cas9 function or dCas9-effector tethering[7]. Because dCas9 binds without discrimination all sgRNAs regardless of their cognate target, such approaches cannot be easily scaled up to implement orthogonal transcriptional programs across multiple genes. While Cas9 variants with divergent protospacer adjacent motif specificities can provide an orthogonal framework for CRISPR-TRs[8], their utility in the design of inducible systems is mitigated by the necessity of extensive protein engineering and the metabolic costs associated with protein delivery.

To address these limitations, we have developed a versatile inducible-CRISPR-TR platform based on minimal engineering of the sgRNA. This system relies on appending native sgRNAs with a spacer-blocking hairpin (SBH) structure, which efficiently silences CRISPR-TR activity. Using this core principle, we devised a range of inducible SBH (iSBH) modules by grafting various RNA-cleaving units on the resulting stem (Fig. 1b). We show that iSBH-mediated conditional regulation of quiescent sgRNAs displays virtually no detectable OFF-state activity, and can be controlled by both proteins (RNA endonucleases) and single-stranded DNA oligonucleotides. Finally, using this platform, we demonstrate highly specific parallel activation of multiple genes using a single inducer and independent control with orthogonal inducer/gene target pairs.

## Results

**Design and optimization of SBH-sgRNAs**. The *Sp*Cas9 sgRNA is composed of a 20-nucleotides (nt) spacer sequence complementary to the target DNA, followed by an ∼80 nt trans-activating crRNA scaffold (tracrRNA)[9,10]. Binding to specific DNA targets occurs when the sgRNA–Cas9 complex encounters a spacer-matching sequence upstream of an NGG protospacer adjacent motif[11]. Since this process is dependent on Watson–Crick base pairing, we reasoned that appending a spacer-complementary 'back-fold' extension at the 5′ end of the sgRNA would generate a 'spacer blocking hairpin', thus effectively silencing CRISPR-TR activity (Fig. 1b). To evaluate the potential of SBH-based systems to control CRISPR-TR activity, we first adopted a recently developed reporter assay, which displays potent transgene activation using a single sgRNA[12] (Supplementary Fig. 1). This system relies on targeting a dCas9-effector fusion protein (dCas9-VP64) to an 'enhancer-like' region containing $8 \times$ CRISPR target site (CTS) repeats placed upstream of a fluorescent reporter gene (Supplementary Fig. 1). Experiments comparing native sgRNAs (nv-CTS) with corresponding SBH-sgRNAs containing back-fold extensions covering the entire spacer segment (0 free spacer nucleotides, $SBH^{(0)}CTS$, Fig. 1c) revealed that SBHs fully abrogate CRISPR-TR activity irrespective of the guide spacer sequence (CTS1 or CTS2) (Fig. 1d). To validate back-fold/spacer base pairing as the cause of silencing, we designed control constructs recapitulating the length and/or structure of the SBH extension without pairing to spacer nucleotides. All control SBH-sgRNAs (offset 10 bp hairpin, $SBH^{(ctrl-1)}CTS$; offset 20 bp hairpin, $SBH^{(ctrl-2)}CTS$; scrambled back-fold extension, $SBH^{(ctrl-3)}CTS$) displayed reporter activation for both CTS1 and CTS2 spacers (Fig. 1d and Supplementary Fig. 2a,b).

The strong SBH-mediated repression and its modular architecture provide an ideal framework for evolving inducible systems (iSBH) by replacing the connecting loop with interchangeable sensor-actuator cleaving units (Fig. 1b). To thermodynamically favour back-fold removal post cleavage while maintaining full silencing in the OFF-state, we designed and tested bulged SBH structures (Supplementary Fig. 3a). Insertion of two 2 nt bulges in the stem ($SBH^{(0B)}CTS1$, where 0 = complete spacer coverage and B = bulge stem) maintained full CRISPR-TR inhibition while increasing the predicted structural free energy from $-38.4$ to $-23.7 \, \mathrm{kcal \, mol^{-1}}$ (Supplementary Fig. 3b,c). In contrast, the presence of an additional basal bulge ($SBH^{(0B*)}CTS1$; $G = -15.0 \, \mathrm{kcal \, mol^{-1}}$) destabilized the stem leading to loss of SBH-mediated silencing (Supplementary Fig. 3b,c). Therefore, the $SBH^{(0B)}CTS$ design was used as default stem for subsequent iSBH implementations.

**Protein-mediated activation of iSBH-sgRNAs**. To demonstrate conditional CRISPR-TR activation, we first adapted the SBH platform to couple the transcriptional output of target genes with protein-based inducers. To this end, we engineered an iSBH responsive to the *Pseudomonas aeruginosa* Csy4 endoribonuclease[13] by grafting its cognate RNA motif onto $SBH^{(0B)}CTS$ ($iSBH^{(0B)}Csy4^{(full)}CTS1$) (Fig. 2a,b). As expected, CRISPR-TR was completely silenced in the OFF-state (decoy empty plasmid), while analysis of ON-state reporter expression revealed robust Csy4-mediated CRISPR-TR activation (Fig. 2c). Confirming the specificity of this effect, a single base pair change in the Csy4 recognition sequence ($iSBH^{(0B)}Csy4m^{(full)}CTS1$), previously reported to prevent cleavage[13], rendered the iSBH system insensitive to induction (Fig. 2b,c).

We next performed an iterative optimization of the iSBH design, which aimed to further lower the stem separation free energy and reduce the number of unstructured 5′ residual nucleotides not bound by dCas9 following Csy4 cleavage. This was accomplished by fusing the Csy4 RNA motif with either the distal or proximal $SBH^{(0B)}CTS$ bulge (Fig. 2d and Supplementary Fig. 4a). The resulting designs, $iSBH^{(0B)}Csy4^{(medium)}CTS1$ and $iSBH^{(0B)}Csy4^{(nano)}CTS1$, had a predicted decrease in stem stability and, correspondingly, displayed an increase in both the number of reporter expressing cells and the levels of fluorescence in the presence of Csy4 (ON-state) (Fig. 2e). These two parameters (% activated cells and reporter fluorescence intensity) were then integrated into an arbitrary activation score to facilitate direct comparison between design iterations (see Methods). This analysis revealed an increase in ON/OFF fold changes from 3.2e2 observed for $iSBH^{(0B)}Csy4^{(full)}CTS1$ to 5.6e2 and 1.9e3 for the medium and nano designs, respectively (Fig. 2f). A similar effect was also observed for CTS2 targeting sgRNAs

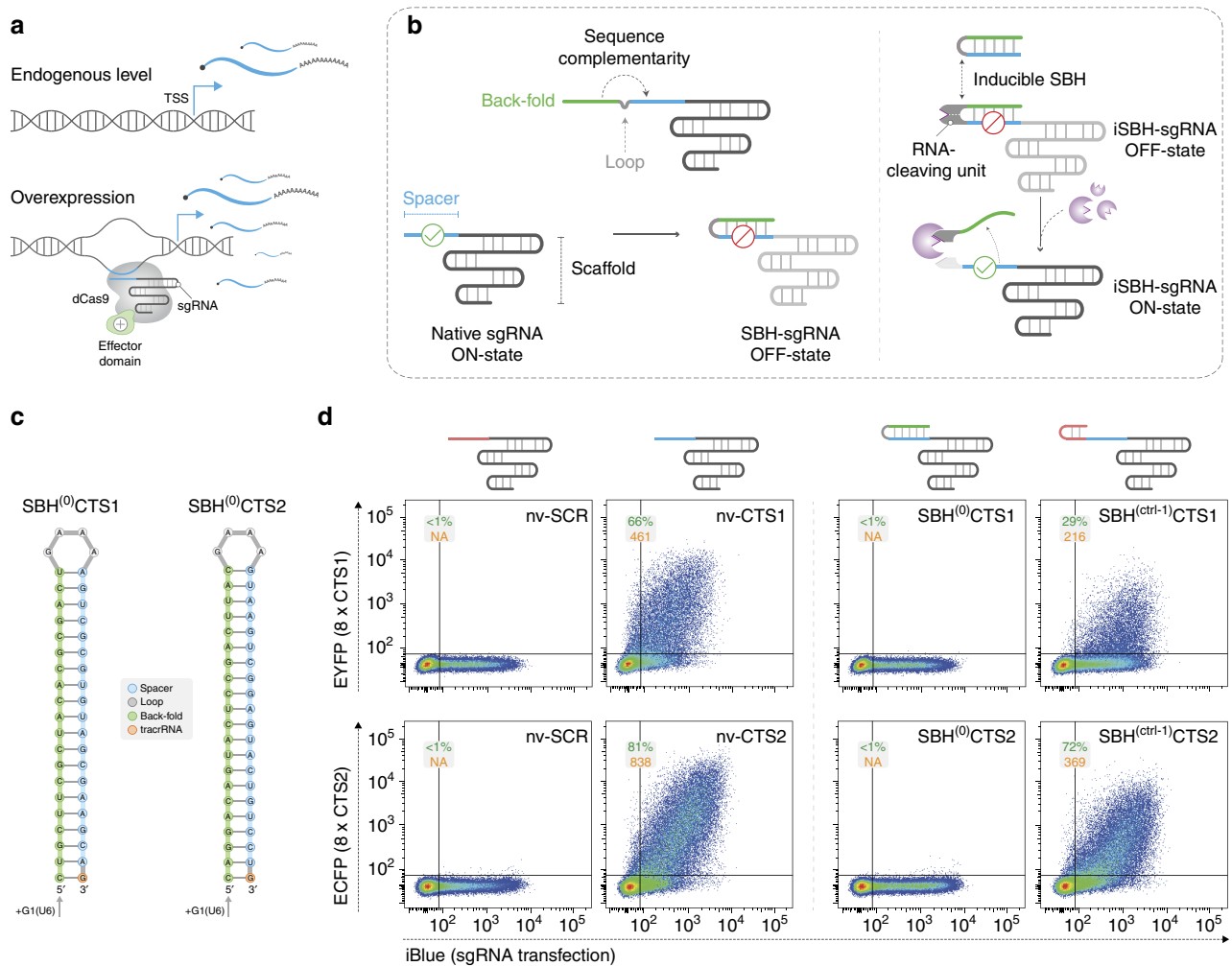

**Figure 1 | Inhibition of CRISPR-TR activity by SBH-sgRNAs. (a)** Schematic representation of CRISPR-TR-based transcriptional modulation.
**(b)** Conceptual framework underlying the design of inducible sgRNAs for the control of CRISPR-TR activity. Appending a back-fold extension to the 5′ end of the native sgRNA promotes the formation of a spacer blocking hairpin (SBH) expected to switch the sgRNA to a quiescent state (OFF-state) (left). Replacing the basic loop with conditional RNA-cleaving units enables generation of inducible SBH designs (iSBH), which can restore CRISPR-TR activity in the presence of specific inducers (spacer release) (right). **(c)** Sequence and secondary structure of prototype SBH designed to silence sgRNAs with spacer targeting CTS1 (left) and CTS2 (right). Superscript annotation ((0)) denotes the number of free spacer nucleotides. +G1(U6) refers to the G nucleotide required for U6 transcription. **(d)** HEK-293T cells were co-transfected with dCas9-VP64, reporter 8xCTS-mCMVp-EYFP and the following sgRNAs: nv-SCR (native sgRNA with scramble spacer sequence); nv-CTS (native sgRNA targeting CTS1 or CTS2); SBH$^{(0)}$CTS (SBH-sgRNAs with full CTS spacer coverage); and SBH$^{(ctrl-1)}$CTS (control SBH-sgRNAs with accessible CTS spacers and offset 5′ end 10 nt hairpin structure). Flow cytometric analysis (48 h post transfection) revealed complete SBH-mediated inhibition of CRISPR-TR activity relative to native and control sgRNAs (see also Supplementary Fig. 2b). Representative flow cytometry scatter plots show reporter activation (EYFP, ECFP channel) plotted against sgRNA transfection (iBlue channel). Flow cytometry plot insets display % of activated cells (double iBlue$^{+ve}$ and EXFP$^{+ve}$, green) and median reporter fluorescence intensity for this population (orange).

(Supplementary Fig. 4b,c) and was consistent with the observation that shorter sgRNA spacers (to 10 nt) can promote strong CRISPR-TR activation (Supplementary Fig. 5a–c)[14]. These results demonstrate that endoribonucleases are effective iSBH-sgRNA inducers that could be genetically encoded to create pre-programmed synthetic circuits in living cells.

**Control of CRISPR-TRs by ASO-responsive iSBH-sgRNAs.** To expand the scope of the iSBH toolkit, we then sought to engineer spacer release mechanisms responsive to short antisense oligonucleotides (ASOs), thus providing a means for temporal exogenous control of CRISPR-TR. Conceptually, this strategy relies on the ability of single-stranded DNA ASOs to bind

complementary iSBH-sensing loops and engage nuclear RNase-H-mediated cleavage of the RNA strand in the resulting DNA/RNA hybrid[15], thus releasing back-fold-mediated CRISPR-TR silencing (Fig. 3a). ASOs are particularly attractive inducers since they have recently emerged as a highly versatile class of compounds that can be safely and efficiently delivered in both cells and organisms to alter gene expression and interfere with post-transcriptional RNA processing[16,17]. In addition, we reasoned that the sequence diversity available for ASO designs would supply an extensive repertoire of possible inducer/target combinations. To establish the feasibility of this approach, ASO inducers were delivered 24 h following transfection of core system components (dCas9-VP64, sgRNA, reporter), and CRISPR-TR-induced reporter expression was assessed 1 day later (Fig. 3a).

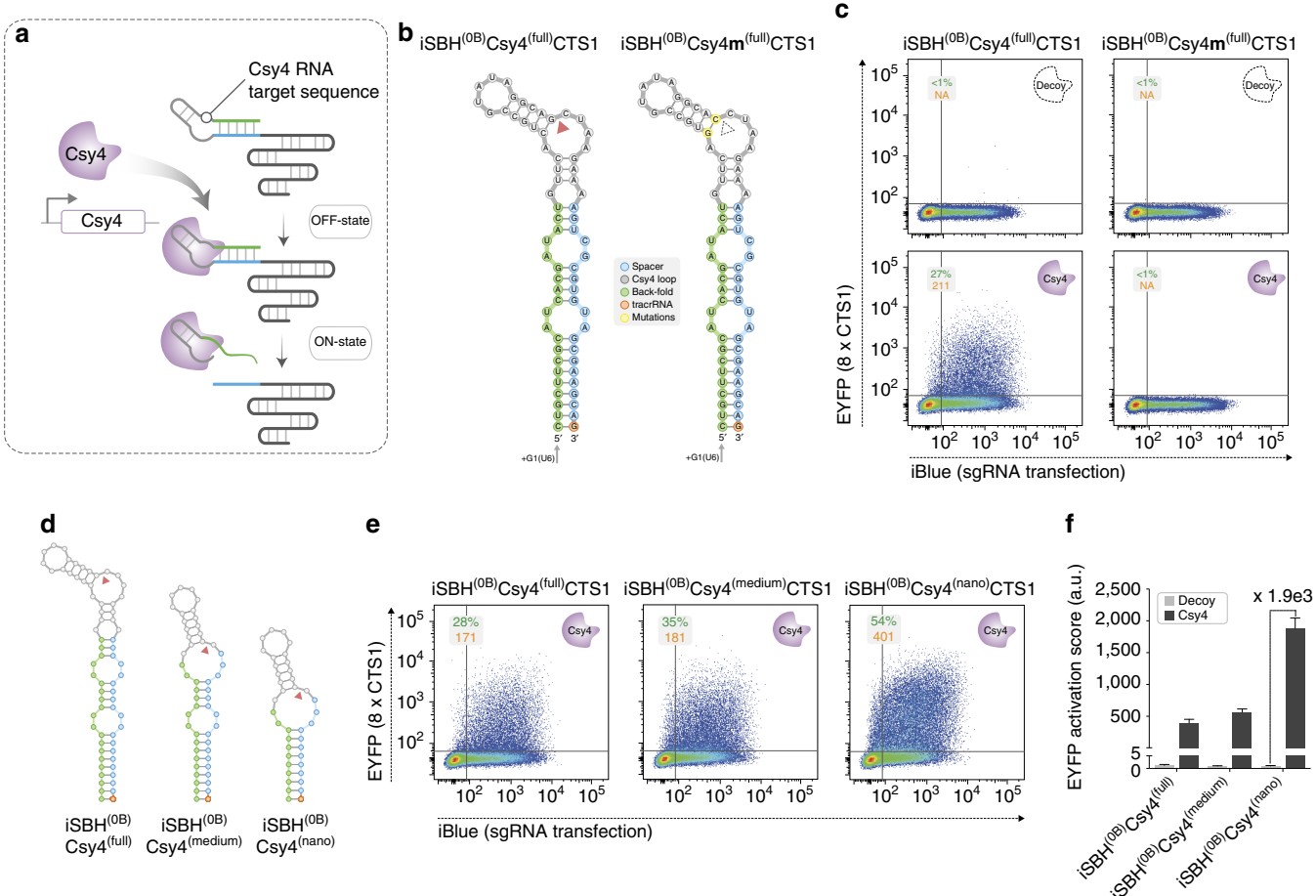

**Figure 2 | Design and optimization of protein-responsive iSBH-sgRNAs.** (**a**) Conceptual framework underlying conditional spacer release using genetically encoded inducers (endoribonucleases). Grafting the Csy4 RNA motif onto the SBH stem allows OFF- to ON-state transition in the presence of the CRISPR-associated endoribonuclease Csy4. (**b**) Sequence and RNA secondary structure of the Csy4-responsive iSBH(0B)Csy4(full)CTS1 and corresponding control mutant variant iSBH(0B)Csy4**m**(full)CTS1 (base pair change (yellow) renders the recognition sequence insensitive to Csy4 cleavage). Red arrow indicates Csy4 cleavage site. (**c**) Representative flow cytometry scatter plots (EYFP reporter fluorescence against iBlue sgRNA transfection) reveal complete silencing in the absence of inducer (decoy = empty plasmid). Robust reporter activation observed in the presence of Csy4 is lost when mutating Csy4-iSBH. (**d**–**f**) Optimization of Csy4-iSBH designs. RNA secondary structures (CTS1 spacer; red arrow Csy4 cleavage site) (**d**) and representative CRISPR-TR assay flow cytometry scatter plots (+ Csy4 ON-state) (**e**) for iSBH(0B)Csy4 full, medium and nano stems. Quantification of EYFP activation score (see Methods) using the three iSBH variants in the presence of a decoy plasmid or Csy4 inducer from three biological replicates (n = 3, mean ± s.d.; a.u., arbitrary units) (**f**). Flow cytometry plot insets display % of activated cells (double iBlue+ve and EYFP+ve, green) and median reporter fluorescence intensity for this population (orange).

Previous studies have shown that ASO-mediated RNase-H cleavage efficiency positively correlates with target site accessibility[18,19]. Based on these considerations, ASO-responsive iSBH-sgRNAs were designed to limit structural interactions within the sensing domain, thus constraining the loop in an open conformation. To create an ASO-responsive iSBH-sgRNA we grafted a 14 nt ASO-sensing loop onto the SBH(0B)CTS (Fig. 3b, iSBH(0B)ASOα-CTS2). As expected, this construct retained full OFF-state silencing in the presence of a decoy scrambled ASO (Fig. 3b). Demonstrating conditional CRISPR-TR activation, delivery of a 14 nt ASO complementary to the sensing-loop (ASOα-14) rendered a 30-fold increase in reporter activation score (Fig. 3c,d). Further extension of ASO length and hybridization footprint aimed to favour strand separation revealed a substantial increase in ON-state CRISPR-TR activity with a 20 nt ASO (ASOα; 113-fold change), while a 25 nt ASO provided a more moderate gain (ASOα-25; 81-fold change) (Fig. 3c,d). Consistent with the effect observed when using a decoy ASO, control experiments employing a scrambled sensing

loop (iSBH(0B)ASO**m**-CTS2) rendered the system insensitive to the inducer, validating the specificity of the ASO-sensing loop interaction (Fig. 3e,f). Similar results were obtained when applying the same design rationale to an iSBH-sgRNA with different spacer (CTS1) and sensing loop sequences (Fig. 3e,f). Interestingly, in contrast to Csy4-iSBH designs, fusing the sensing loop to the SBH(0B)CTS distal bulge reduced CRISPR-TR activity, presumably due to dCas9 interfering with ASO/sensing loop hybridization (Supplementary Fig. 6a–c).

**Implementation of protein-responsive gene modules.** Complex synthetic gene circuits can in principle be reduced to two fundamental gene network modules: (1) branching module whereby a single upstream event simultaneously controls the activity of multiple downstream nodes; (2) orthogonal module which allows asynchronous control of downstream targets using independent inducer/gene pairs (Fig. 4a). Leveraging the versatility and simplicity of the iSBH design, we next sought to

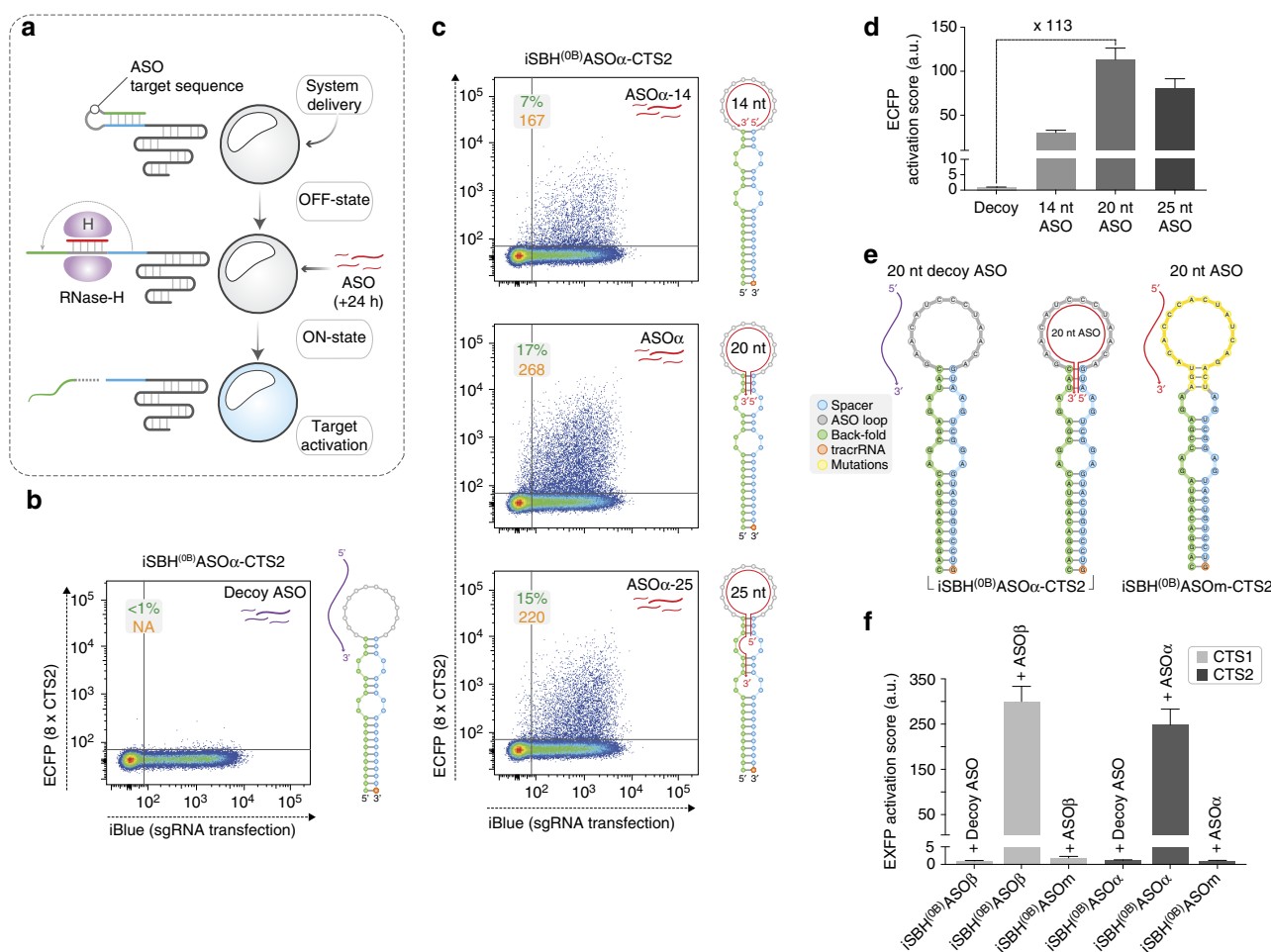

**Figure 3 | Design and optimization of ASO-responsive iSBH-sgRNAs.** (**a**) Implementation of ASO-responsive iSBH designs for temporal control of CRISPR-TR activity. This platform enables delayed sgRNA activation by means of externally delivered cognate ASOs. (**b**) CRISPR-TR assay (ECFP reporter expression flow cytometry) in the presence of iSBH(OB)ASOα-CTS2 and a decoy ASO with scrambled sequence (purple, incompatible base pairing with the sensing loop). (**c**,**d**) Impact of ASO length on the activation of iSBH(OB)ASOα-CTS2 sgRNA (ON-state). Representative flow cytometry scatter plots and corresponding ASO/iSBH pairing diagrams using 14-, 20- and 25-nucleotide (nt) long ASO inducers (red) (**c**). Quantification of ON-state reporter activation score for the conditions shown in **b**,**c** ($n = 3$, mean ± s.d.; a.u., arbitrary units) (**d**). (**e**,**f**) Analysis of ON-state activation and OFF-state inhibition using the optimal 20 nt ASO inducer design on two target genes (CTS1-EYFP and CTS2-ECFP). For each CTS three experimental conditions were compared: decoy ASO and iSBH-sgRNA; matching ASO and iSBH-sgRNA; and matching ASO and iSBH-sgRNA (non-matching sensing loop) (**e**). Quantification of reporter activation scores from three biological replicates for each condition. No activation above background was detected in control conditions (decoy ASO or matching ASO + iSBH(0B)ASO**m**-CTS), while robust CRISPR-TR-mediated reporter expression was elicited by the presence of active ASOs (**f**). Flow cytometry plot insets show % of activated cells (double iBlue+ve and ECFP+ve, green) and median reporter fluorescence intensity for this population (orange).

establish its potential as a framework for the assembly of gene networks. The implementation of branching and orthogonal control can be assessed for both protein- and ASO-responsive iSBH designs using a dual reported system ([8 × CTS1-EYFP] − [8 × CTS2-ECFP]). Furthermore, we adapted the synergistic activation mediator (SAM) system[6] to demonstrate the ability of iSBH-sgRNAs to programme conditional activation of endogenous genes (Supplementary Fig. 7a,b).

To assemble a branching module using protein-responsive iSBH-sgRNAs and demonstrate simultaneous activation of multiple targets conditioned on the presence of Csy4, we first designed iSBH(OB)Csy4(nano)CTS1 and iSBH(OB)Csy4(nano)CTS2 sgRNAs and co-transfected them along with dCas9-VP64 and the dual reporter system. In both cases, robust CRISPR-TR-mediated parallel expression was observed in the presence of Csy4, while no detectable activation was noticed in the absence of inducer

(Fig. 4b and Supplementary Fig. 8a). Furthermore, sequentially scrambling the spacer sequence of each iSBH-sgRNA resulted in the expression of a single target gene, validating the dependence of branched activation on the presence of both guides (Fig. 4b). Applying the same design framework, we then engineered SAM sgRNAs (containing two MS2 loops) to accommodate Csy4-responsive iSBHs and programme parallel conditional activation of endogenous *HBG1* and *IL1B* genes[6]. Delivery of the resulting constructs (iSBH(OB)SAM-Csy4(nano)HBG1 and iSBH(OB)SAM-Csy4(nano)IL1B, respectively) to HEK293T cells, along with the SAM system, showed concurrent upregulation of both genes relative to endogenous levels in the presence of Csy4 (Fig. 4c).

To design iSBH-based orthogonal gene modules implementing independent protein inducer/target pairs, we next created full, medium and nano iSBH-sgRNAs responsive to the Cas6A endoribonuclease form *Thermus thermophilus*[20] (Supplementary

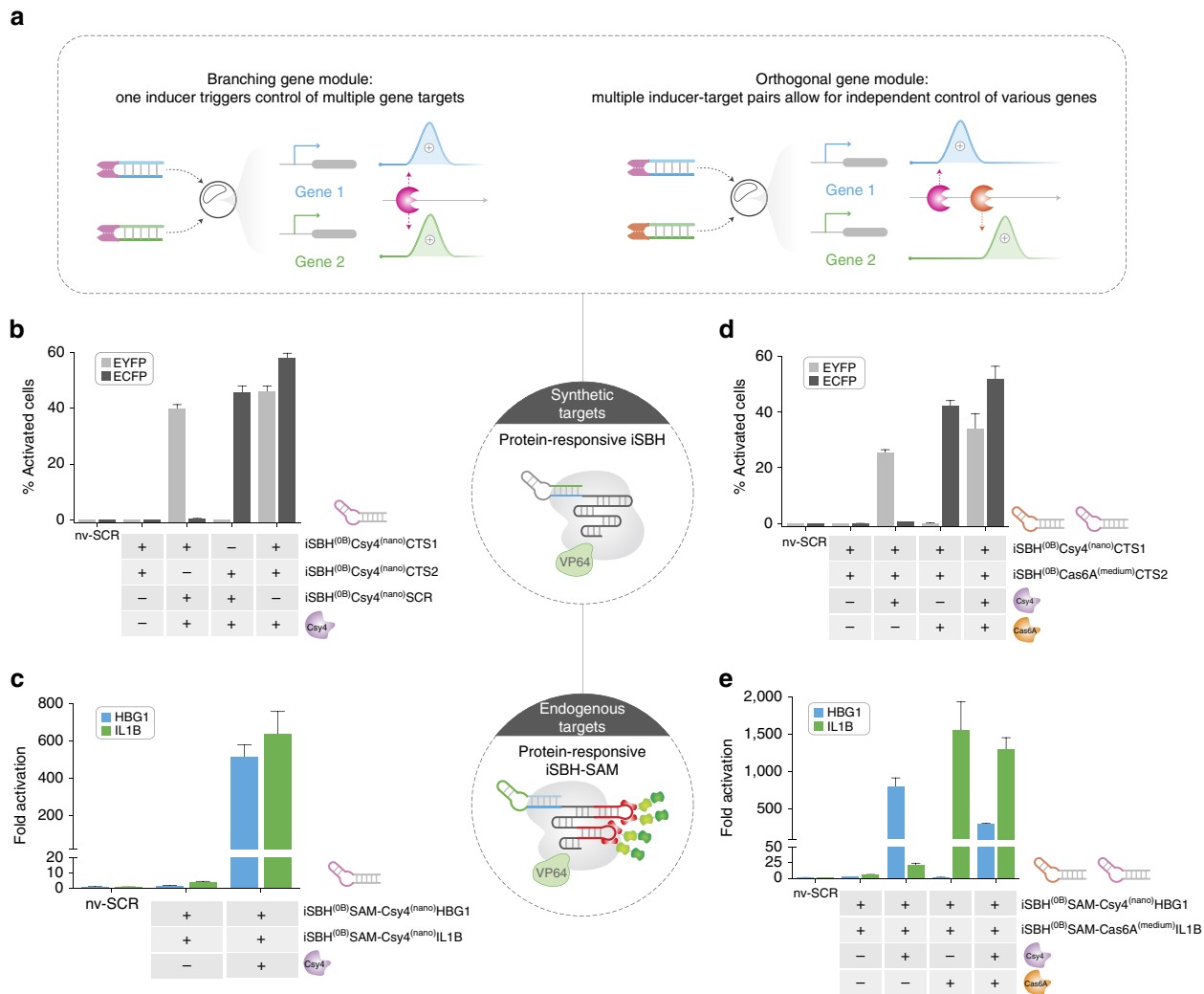

**Figure 4 | Assembly of gene network modules with protein-responsive iSBH sgRNAs.** (**a**) Schematic representation of branching and orthogonal gene network modules using iSBH-sgRNAs. iSBH-sgRNAs programmed with specific sensing loops (purple and orange) and/or spacers (blue and green) enable rapid generation of parallel and orthogonal inducer/target gene pairs, facilitating synchronous or asynchronous control of transcriptional programs. (**b**) Concurrent activation of two reporter genes using protein-responsive iSBH-sgRNAs (branching module). Csy4-responsive iSBH(OB)Csy4(nano)CTS1 and iSBH(OB)Csy4(nano)CTS2 were co-transfected in the absence or presence of the inducer. To confirm Csy4-mediated specific activation of CTS1 and CTS2 target genes, each corresponding iSBH-sgRNA was co-transfected with a control iSBH-sgRNA carrying a scramble spacer (iSBH(OB)Csy4(nano)SCR). (**c**) Concurrent overexpression of two endogenous genes (*HBG1* and *IL1B*) using Csy4-responsive iSBH-sgRNAs (branching module). Co-transfection of iSBH(OB)SAM-Csy4(nano)HBG1, iSBH(OB)SAM-Csy4(nano)IL1B elicits an increase in the corresponding genes transcript levels in the presence of Csy4 compared with nv-SCR or decoy inducer controls. (**d**) Orthogonal activation of two target genes using protein-responsive iSBH-sgRNAs (orthogonal module). Csy4- and Cas6A-responsive iSBH(OB)Csy4(nano)CTS1 and iSBH(OB)Cas6A(medium)CTS2, respectively, were co-transfected in the absence of any inducer, the presence of each individual inducer or a combination of the two. (**e**) Wiring of *HBG1* and *IL1B* gene output with independent inducers (Csy4 and Cas6A, respectively). Coexpression of iSBH(OB)SAM-Csy4(nano)HBG1 and iSBH(OB)SAM-Cas6A(medium)IL1B leads to inducer-specific orthogonal regulation of transcriptional gene output compared with nv-SCR control. For (**b**,**d**) graphs show percentage of activated cells (EYFP and/or ECFP positive) among the entire sgRNA-transfected population (iBlue positive) ($n = 3$ biological replicates for each condition; mean ± s.d.). Quantification of cell fractions expressing EYFP, ECFP or both EYFP/ECFP relevant to conditions in **b**,**d** is displayed in Supplementary Fig. 8. nv-SCR refers to a control native sgRNA with a scramble spacer sequence. dCas9-VP64 and a dual-reporter system plasmid ([8 × CTS1-EYFP] − [8 × CTS2-ECFP]) were co-transfected by default in all conditions. For (**c**,**e**) data displays fold change in transcript levels measured by RT-qPCR ($n = 3$ biological replicates ( × 3 technical replicates), mean ± s.d.). Cas9-VP64 and MCP-p65-HSF1 were co-transfected by default in all conditions.

Fig. 9a). Similar to Csy4-iSBH-sgRNAs, Cas6A-responsive hairpins showed full CRISPR-TR silencing in the OFF-state and robust ON-state target gene activation (Supplementary Fig. 9b,c). Optimal iSBH(OB)Csy4 and iSBH(OB)Cas6A designs (best ON/OFF characteristics) were then selected to condition the expression of synthetic (EYFP, ECFP) and endogenous (*HBG1*, *IL1B*) targets on the presence of Csy4 and Cas6A. Demonstrating successful implementation of an orthogonal module, each target gene was exclusively activated by its corresponding

trigger (Csy4 or Cas6A), with no detectable crosstalk between independent branches (Fig. 4d,e). As expected, simultaneous expression of both genes was achieved when Csy4 and Cas6A were co-delivered (Fig. 4d,e, Supplementary Fig. 8b).

**Implementation of ASO-responsive gene modules.** We next set out to implement corresponding branching and orthogonal modules exogenously controlled by ASO triggers. ASO-mediated

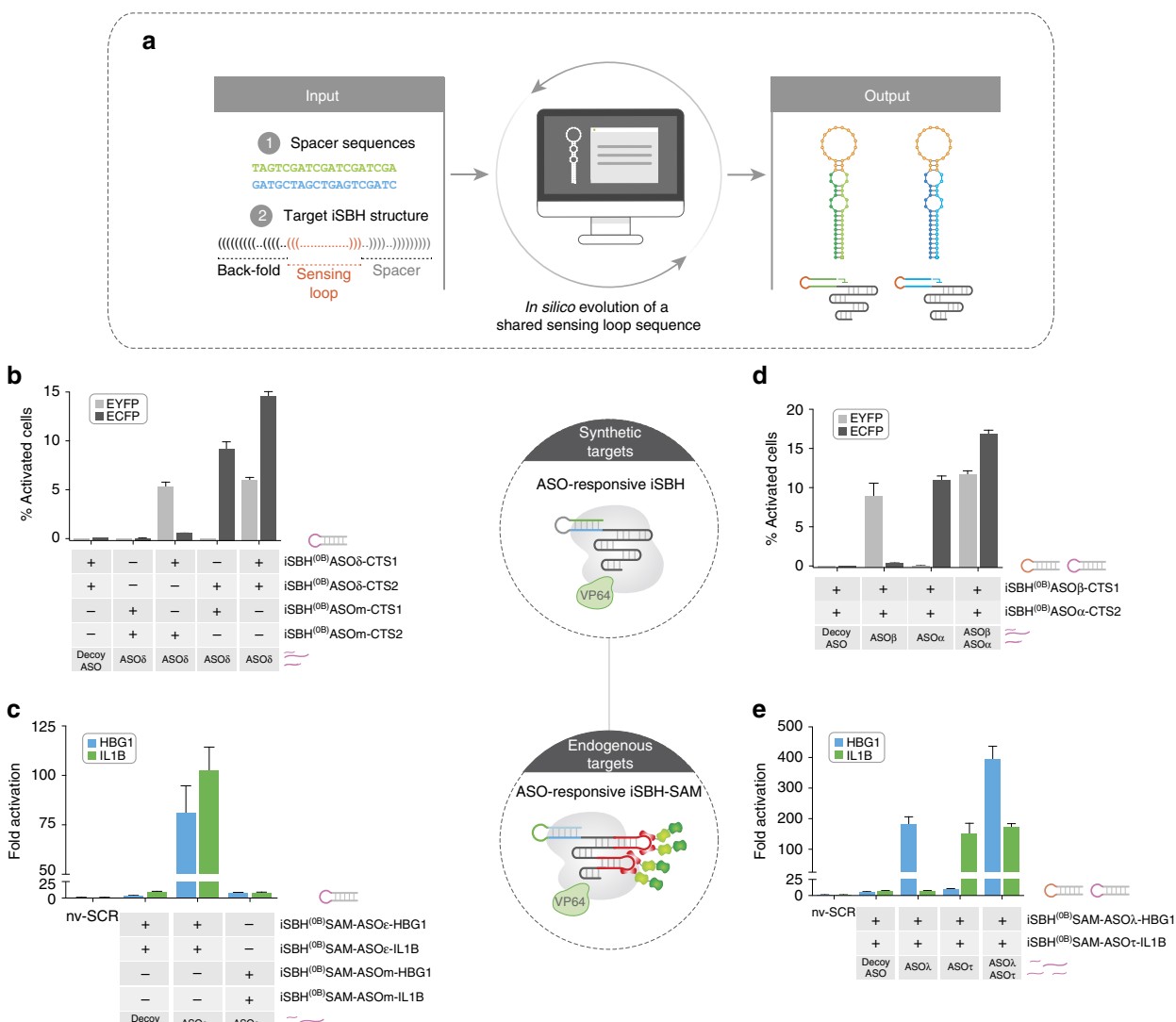

**Figure 5 | Assembly of gene network modules with ASO-responsive iSBH sgRNAs.** (**a**) Schematic of the *iSBHfold* algorithm. Based on user-specified spacer sequences and desired iSBH secondary structure, the software employs a genetic algorithm to evolve a pool of potential ASO sensing loops. The resulting sequences are compatible with optimal iSBH structures across multiple spacers. (**b**) Parallel activation of target genes using ASO-responsive iSBH-sgRNAs (branching module). iSBH$^{(OB)}$ASOδ-CTS1 and iSBH(0B)ASOδ-CTS2 containing a shared sensing loop were co-transfected with dCas9-VP64 and the dual-reporter system. Decoy ASO or trigger ASOδ were delivered to cells 24 h post transfection. Parallel experiments using iSBH-sgRNAs with mutant sensing loops (iSBH$^{(OB)}$ASOm-CTS1 and iSBH$^{(OB)}$ASOm-CTS2) were carried out to confirm the specificity of the observed effects. (**c**) Conditional overexpression of *HBG1* and *IL1B* using a single ASO. iSBH$^{(OB)}$SAM-ASOε-HBG1 and iSBH$^{(OB)}$SAM-ASOε-IL1B responsive to a shared trigger ASOε were used to implement a branching module. Delivery of ASOε at 24 h post transfection resulted in an increase in transcript levels for both genes compared with nv-SCR control, a decoy ASO or iSBH-sgRNAs containing mutant sensing loops (iSBH$^{(OB)}$SAM-ASOm-HBG1, iSBH$^{(OB)}$SAM-ASOm-IL1B). (**d**) Implementation of an orthogonal gene activation module using ASO-responsive iSBH-sgRNAs. iSBH$^{(OB)}$ASOβ-CTS1 and iSBH$^{(OB)}$ASOα-CTS2 containing distinct sensing loop were supplemented 24 h post transfection with a decoy ASO, ASOβ, ASOα or a combination of ASOβ + ASOα. (**e**) iSBH$^{(OB)}$SAM-ASOλ-HBG1 and iSBH$^{(OB)}$SAM-ASOτ-IL1B containing different ASO-sensing loops were supplemented with decoy ASO, ASOλ, ASOτ or a combination of ASOλ + ASOτ. Orthogonal regulation of *HBG1* and *IL1B* transcription was observed in the presence of matching ASOs compared with a control nv-SCR sgRNA. For (**b,d**) n = 3 biological replicates for each condition (mean ± s.d.). Quantification of cell fractions expressing EYFP, ECFP or both EYFP and ECFP relevant to conditions in **b,d** is displayed in Supplementary Fig. 8. For (**c,e**) n = 3 biological replicates ( × 3 technical replicates) (mean ± s.d.).

branching requires the evolution of a shared sensing loop, which should display optimal folding properties (accessibility to ASO pairing) across multiple iSBH-sgRNA spacer sequences. To automate this process we have created *iSBHfold* (http://apps.molbiol.ox.ac.uk/iSBHfold/cgi-bin/iSBHfold.cgi), a custom software combining genetic algorithm with RNA secondary structure predictions[21] (see Supplementary Software). Using this software we engineered iSBH$^{(OB)}$ASO-sgRNAs

(ASOδ inducer) and iSBH$^{(OB)}$SAM-ASO-sgRNAs (ASOε inducer) targeting synthetic (EYFP, ECFP) and endogenous (*HBG1*, *IL1B*) gene pairs, respectively (Fig. 5a and Supplementary Fig. 10). The corresponding modules displayed ON-state branching behaviour following delivery (24 h post transfection) of their cognate inducer ASO, and complete silencing in the presence of decoy ASOs (scramble sequence) (Fig. 5b,c and Supplementary Fig. 8c). Furthermore, control experiments aimed

to decouple each target gene from the inducer by sequentially mutating the sensing loops revealed the expected loss of gene activation in the corresponding branch (Fig. 5b).

The availability of a broad inducer pool for ASO-responsive iSBH-sgRNAs provides an optimal framework for the construction of CRISPR-TR-based orthogonal gene modules in mammalian cells. To illustrate this potential, we have generated a series of individual ASO/iSBH pairs coupling distinct ASO inducers with conditional activation of either synthetic targets (ASOβ/EYFP, ASOα/ECFP) or endogenous genes (ASOλ/HBG1, ASOτ/IL1B). Temporal induction of quiescent iSBH(0B)ASO or iSBH(0B)SAM-ASO sgRNAs with separate or simultaneous ASO delivery resulted in the anticipated target gene activation profiles without any apparent interference between individual branches (Fig. 5d,e and Supplementary Fig. 8d). Together, these results demonstrate the relevance of the iSBH framework in facilitating assembly of basic modules for construction of synthetic gene circuits.

**Ribozyme-mediated activation of SBH-sgRNAs.** In addition to the rich repertoire of genetically encoded and externally delivered inducers provided by nucleases and ASOs, iSBH-sensing modules could be evolved to respond to other categories of ligands using self-cleaving allosteric hammerhead ribozymes (aHHRz) (Supplementary Fig. 11a)[22]. Previous studies have shown that aHHRz can be effectively used for the construction of ligand-controlled synthetic circuits[22–24]. To establish the feasibility of leveraging the HHRz design as a self-contained spacer release mechanism, we fused the HHRz structure onto the SBH(0B)CTS scaffold (Supplementary Fig. 11b). Comparative analysis of SBH-sgRNAs containing catalytically active HHRz (SBH(0B)HHRz-CTS1) or inactive HHRz (SBH(0B)mHHRz-CTS1) demonstrated robust HHRz-mediated activation of reporter gene expression and complete silencing in the OFF-state (Supplementary Fig. 11c). Future iterations of this generic HHRz-SBH scaffold could take advantage of established in vivo or in vitro RNA aptamer evolution strategies to engineer iSBHs responsive to a variety of protein, nucleotide and small-molecule ligands[25–28].

## Discussion

The advantage of using sgRNA-based inducible systems for synthetic biology applications has been recently showcased in a study demonstrating the ability to rewire cellular pathways by CRISPR-TR with modified sgRNAs containing ligand-responsive riboswitches[29]. These sgRNA 'signal conductors' employ a strand-displacement mechanism to transition between OFF and ON states and can be coupled to a variety of inducers and dCas9 effectors. Using a distinct spacer release mechanism, the iSBH platform offers a versatile and simple 'plug and play' alternative solution for accurate conditional activation of CRISPR-based systems in eukaryotic cells. Since this strategy relies on minimal alterations of the sgRNA scaffold, iSBH-based inducibility is in principle compatible with all CRISPR derivatives including, genome editing[30], genetic and epigenetic alteration[31], base editing[32] and labelling of genomic loci[33]. Notably, with regard to the assembly of GNs, the integration of iSBH with previous effector-binding sgRNA scaffolds[6,34,35] provides an opportunity to encode within a single RNA molecule a complete transcriptional program. Inherent to its highly versatile modular design, the system could thus be adapted to a variety of inducers (iSBH spacer release mechanism), target genes (spacer identity) and transcriptional outputs (effector domain tethering). Based on these considerations, we envision that the iSBH framework will facilitate the assembly of more complex gene circuits while minimizing the potential for crosstalk between biological parts. Notably, since the iSBH

sgRNAs display nearly complete silencing in the OFF-state, this platform will be particularly suitable for the design of synthetic GNs targeting pro-apoptotic kill switch programs in human cells.

We anticipate that iSBH-based CRISPR systems will also provide a valuable tool for reverse engineering studies aiming to understand transcriptional programs underpinning natural development (for example, cell differentiation) as well as disease progression (such as oncogene dynamics). Tissue-specific expression of iSBH protein inducers (Csy4, Cas6A and so on) and sequential sgRNA activation could be used to facilitate cell labelling and lineage tracing throughout development or in disease states (cancer)[36]. Finally, coupling iSBH (protein-based spacer release mechanisms) with protein inducers under the control of engineered cellular receptors[37] might also enable the design of more complex prosthetic gene networks for research and therapeutic purposes.

## Methods

**Molecular biology.** All cloning DNA oligonucleotides and PCR primers were obtained from Integrated DNA Technologies (IDT) (see Supplementary Tables 1 and 2). Restriction enzymes were purchased from New England Biolabs (NEB) and used according to the manufacturer's protocols. In all cloning experiments, both the linearized DNA vector and the corresponding inserts were purified using the QIAquick Gel Extraction Kit (Qiagen). Subsequently, vectors were dephosphorylated using Antarctic Phosphatase (NEB), and ligations were carried out with T4-DNA ligase (NEB). DH5α-competent cells (custom made) were transformed by heat shock, and cells were grown on LB plates containing selection antibiotics (ampicillin or kanamycin) for 16 h at 37 °C. Single colonies were grown in LB + antibiotic liquid media overnight at 37 °C, and DNA was extracted using the QIAprep Spin Miniprep Kit (Qiagen). All constructs were validated by Sanger sequencing (Eurofins genomics) before transfection in HEK293T cells.

**SBH and iSBH cloning.** First, the coding sequence of iBlue fluorescent protein was PCR amplified from the iBlue-N1 vector (gift from Michael Davidson (Addgene plasmid 54781)) using the Fwd_iBlue and Rev_iBlue_BsrGI primers. The resulting fragment was then assembled with the SV40 promoter by fusion PCR using primers Fwd_SV40_NcoI and Rev_SV40, and cloned between the NcoI and BsrGI sites in the pcDNA3.1 vector to generate pcDNA3.1_SV40-iBlue-pA. Subsequently, an sgRNA cassette containing the U6 promoter, sgRNA scaffold and U6 terminator (Supplementary Fig. 14) was PCR amplified from the pX330 vector (gift from Feng Zhang (Addgene plasmid 42230)) using primers Fwd_sgRNA-cassette_SpeI and Rev_sgRNA-cassette_BcoDI, and cloned into the SpeI and BbsI sites of pcDNA3.1_SV40-iBlue-pA vector. vector. For simplicity, the resulting vector is referred to as 'sgRNA-backbone' and was used for the generation of all native, SBH and iSBH-sgRNAs targeting synthetic CTSs. For native sgRNAs, spacer sequences were synthesized (IDT) and cloned between BbsI sites in the sgRNA-backbone vector as previously[38]. SBH and iSBH-sgRNAs were similarly generated by synthesizing the entire back-fold–loop–spacer sequence flanked by corresponding 5′ and 3′ overhangs (CACC and AAAC, respectively). An exception applied to SBH containing the HHRz-cleaving unit. In this case, the Schistosoma mansoni HHRz sequence[39] was assembled from multiple synthesized fragments (IDT, HHRz_oligo 1–4 and flanking primers) by assembly PCR. Csy4 and Cas6A RNA motifs were adapted from Haurwitz et al.[13] and Niewoehner et al.[20] respectively. ASO sensing loops were evolved using the iSBHfold custom-made algorithm as described below. To generate iSBH-sgRNAs targeting endogenous genes, the same general strategy was used to clone the relevant constructs in the SAM_U6-sgRNA-2xMS2 vector (gift from Feng Zhang (Addgene plasmid 61424)). Prediction of SBH and iSBH secondary structures as well as stem minimum free energy calculations were performed using the NUPACK package (http://www.nupack.org[21]). All SBH and iSBH constructs used in this study are shown in Supplementary Fig. 12 and Supplementary Table 2.

**Inducer cloning and synthesis.** The NLS sequence contained in the pX330 vector was PCR amplified (Fwd_pX330_NLS and Rev_pX330_NLS) and cloned between HindIII and NheI sites upstream of Pseudomonas aeruginosa Csy4 (human codon optimized) in the PGK1p-Csy4-pA vector (gift from Timothy Lu (Addgene plasmid 55196)). The resulting construct (PGK1p-NLS-Csy4-pA) was used to express Csy4 under the PGK promoter. The sequence of Thermus thermophillius Cas6A was obtained from the NCBI (National Center for Biotechnology Information; accession no. TTHA0078), human codon optimized (https://www.idtdna.com/CodonOpt, Supplementary Fig. 14), and cloned as a gBlock fragment (IDT, see NLS-Cas6A sequence below) instead of Csy4 in the PGK1p-NLS-Csy4-pA using NheI and NotI. All ASO sequences used in this study are listed in Supplementary Table 3. To improve stability within the cellular

environment, all ASOs were synthesized using a phosphodiester backbone and three or four terminal phosphorothioate bonds(*) (IDT).

**HEK293T transfections.** Low-passage mycoplasma tested HEK293T cells (gift from Professor Ahmed Ashour Ahmed, clone number ATCC-CRL-11268) were thawed and passaged for 1 week before use. To assess transfection efficiency, each batch was tested using a reference EGFP-expressing plasmid (pcDNA3.1_CMVp-EGFP-pA). HEK293T cells were cultured in Dulbecco's modified Eagle's medium (Thermo-Fisher Scientific) supplemented with 15% fetal bovine serum and 1% penicillin–streptomycin (full media) at 37 °C and 5% $CO_2$, and passaged every 48 h in a 1:6 ratio for ∼2 months before being replaced with a new batch. For transfection experiments, cells were seeded in 12-well plates and transfected 24 h later at ∼70% confluency. Cells were transfected with polyethylenimine (PEI; Sigma-Aldrich, 1 mg ml$^{-1}$) as previously described[40]. Briefly, full media were removed and replaced by transfection media (Dulbecco's modified Eagle's medium + 2% fetal bovine serum). DNA and PEI were mixed in a 2:3 ratio (μg DNA/μl PEI) in 100 μl Opti-MEM (Thermo-Fisher Scientific), vortexed for 10 s, incubated for 15 min at room temperature and added to cells. At 12 h after transfection, media were changed back to full media, except when ASO inducers were delivered (see below). For each sgRNA class, the following transfection conditions were used:

SBH-sgRNAs: 250 ng of either 8 × CTS1-mCMVp-EYFP-pA (Addgene 55197) or 8 × CTS2-mCMVp-ECFP-pA (gift from Timothy Lu (Addgene plasmid 55197 and 55198)), 250 ng of dCas9-VP64 (gift from George Church (Addgene plasmid 47319)) and 500 ng of sgRNA plasmid of interest containing various native spacers or SBH variants were co-transfected in each well. Cells were collected 48 h after transfection for flow cytometry analysis.

Protein-responsive iSBH-sgRNAs: 500 ng protein-responsive iSBH-sgRNAs plasmids were co-transfected with 250 ng of PGK1p-NLS-Csy4-pA or PGK1p-NLS-Cas6A-pA inducer plasmids in addition to the reagents mentioned above. For simultaneous inducer delivery, 125 ng of each inducer plasmid was used. For control experiments (decoy inducer), 250 ng of backbone pcDNA3.1 vector was transfected instead of inducer plasmids.

ASO-responsive iSBH-sgRNAs: 500 ng ASO-responsive iSBH-sgRNAs were co-transfected along with dCas9-VP64 and reporter constructs (see above). At 24 h post transfection, new transfection media were added to cells and ASOs (100 nM final concentration) were delivered in a solution containing 500 ng carrier DNA (pcDNA3.1) in 100 μl Opti-MEM. Cells were collected 48 h post transfection.

SAM-based iSBH-sgRNAs: SAM_dCas9-VP64 and SAM_MS2-P65-HSF1 (gift from Feng Zhang (Addgene plasmid 61422 and 61423)) were co-transfected along with protein- or ASO-responsive iSBH-SAM-sgRNA plasmids (SAM_U6-sgRNA-2 × MS2 constructs) in a 1:1:1 ratio[6] (500 ng of each plasmid). Similar inducer delivery protocols (concentrations and timing) were used for protein-responsive and ASO-responsive iSBH-SAM-sgRNAs as described above. Cells were collected 48 h post transfection and total RNA was extracted from each well for quantitative reverse transcription–PCR (RT–qPCR) analysis (see below).

The impact of iSBH-sgRNA expression on cellular state was assessed by flow cytometry (live/dead staining) and confocal microscopy (4,6-diamidino-2-phenylindole and F-actin staining), and compared with native sgRNAs. The iSBH-sgRNA-transfected cells displayed normal live/dead cell staining counts, as well as wild-type morphology and nuclear integrity (Supplementary Fig. 13).

**Flow cytometry experiments.** For all CRISPR-TR experiments using fluorescent reporter constructs (8 × CTS1-mCMVp-EYFP-pA or 8 × CTS2-mCMVp-ECFP-pA), media were removed 48 h post transfection and cells were washed with 1 × phosphate buffer saline, trypsinized (0.05% trypsin–EDTA, Thermo Fisher Scientific), collected by centrifugation (3 min at 500g) and kept in phosphate buffer saline on ice. Flow cytometry measurements were carried out within 30–60 min from cell collection on a BD LSR Fortessa Analyzer (BD Biosciences). The same laser intensities and wavelength were used for all experiments. Forward scatter and side scatter were used to isolate healthy singleton cells. For each condition, 1e5 total events were recorded.

**Analysis of flow cytometry data.** To calculate the % activated cells value, events were first partitioned into sgRNA-transfected (iBlue$^{+ve}$) and untransfected (iBlue$^{-ve}$) cells from the parent population (viable single cells). The iBlue$^{+ve}$ population was further gated on the corresponding EXFP channel to isolate inducer-activated cells displaying reporter fluorescence above background (double positive iBlue&EXFP$^{+ve}$). Therefore, % activated cells refers to the activated fraction relative to the entire iBlue$^{+ve}$ cell population. An EXFP activation score (arbitrary units) was then calculated by multiplying the frequency of activated cells in the iBlue$^{+ve}$ population by their median reporter fluorescence according to the formula below[12,41].

$$\% \, EXFP^{+ve}_{iBlue^{+ve}} \times EXFP^{median}_{iBlue\&EXFP^{+ve}}$$

This metric provides a weighted fluorescence value integrating both the spread and strength of activation. It should be noted that the activation score is inherently an arbitrary value and is influenced by the position of the EXFP$^{+ve}$ gate. To avoid zero values and obtain representative fluorescence measurements in the OFF-state

(where very few cells are activated), this gate was set to allow only ∼0.1% false positive in the negative control condition (scramble native sgRNA, nv-SCR). This stringent cutoff was applied to ensure that activation score calculations are sensitive to even minimal OFF-state background activation (for example, even if as little as 1% iBlue$^{+ve}$ cells would display activation in the OFF-state, this will translate into a 10-fold increase in activation score compared with the nv-SCR condition). However, since most iSBH systems fully silence CRISPR-TR in the absence of an inducer (% activated cells ∼0.1%), the activation score fold change between OFF and ON-state can exceed by one order of magnitude the observed ON-state median reporter fluorescence.

**RT-qPCR analysis.** For CRISPR-TR experiments using endogenous targets, cells were collected 48 h post transfection and total RNA was extracted from each well using the RNeasy Mini Kit (Qiagen) according to the manufacturer's protocol. Immediately following RNA extraction, 1 μg total RNA was reverse transcribed (random hexamer priming) using the QuantiTect Reverse Transcription Kit (Qiagen). Quantitative PCR was carried out using the SsoAdvanced Universal SYBR Green Supermix kit (Bio-Rad) on a CFX384 real-time system (Bio-Rad). All forward and reverse primer pairs used for GAPDH, dCas9-VP64, HBG1 and IL1B are listed in Supplementary Table 4.

Data were analysed using the ΔΔCt method. Briefly, for each condition Cas9-VP64 and target gene Ct values were normalized to GAPDH (transcript level for X gene = 2^($Ct_{GAPDH} - Ct_X$)). Before fold-change calculation, HBG1 and IL1B transcript levels were normalized to dCas9-VP64 levels to account for variation in transfection efficiencies. The fold-change increase in transcript levels was calculated by dividing target gene values in iSBH-sgRNA conditions to those in control samples (nv-SCR sgRNA) according to the formula below (e = experiment (iSBH-sgRNA) and c = control (nv-SCR sgRNA)).

$$\text{Fold change} = 2^{\left(Ct^e_{GAPDH} - Ct^e_{HBG1}\right) - \left(Ct^e_{GAPDH} - Ct^e_{dCas9}\right)} \Big/ 2^{\left(Ct^c_{GAPDH} - Ct^c_{HBG1}\right) - \left(Ct^c_{GAPDH} - Ct^c_{dCas9}\right)}$$

**iSBHfold algorithm.** A custom algorithm was developed to automate the evolution of shared ASO sensing loops (ASLs) displaying optimal ASO-iSBH folding across multiple spacer sequences (see Supplementary Fig. 10). The algorithm aims to output an ASL which favours proper folding of ASO-iSBH structures whereby the default SBH$^{(0B)}$ bulge stem architecture is maintained and a 14 nt open loop conformation is available for ASO targeting. Selection of ASL candidates is accomplished by enriching a starting ASL pool containing random sequences using a genetic algorithm. Given p spacer sequences (SP1,…, SPp) the algorithm attempts to identify a set of ASLs which satisfy all structural constraints for ASL-SP1,…,ASL-SPp SBH$^{(0B)}$ hairpins. The system is initialized with an ASL pool comprising randomly generated 20 nt sequences (14 nt ssRNA segment flanked at both 5′ and 3′ ends with three stem-complementary nucleotides) (N = 150 sequences were used in this study). Based on this initial pool the system iterates over several generations. (1) Each sequence from the pool is recombined with a randomly selected partner sequence to produce two offspring sequences which are added to the existing pool. Recombination events are induced at random positions along the 20 nt segment. (2) Additional sequences obtained by randomly mutating existing ASLs from the pool and (3) fully random sequences are then integrated in the pool. (4) Each ASL sequence is then complemented with a back-fold (5′ end) and spacer (3′ end) to obtain the RNA sequences of the corresponding ASL-SP1,…,ASL-SPp SBH$^{(0B)}$ hairpins (p sequences per ASL). For each of these sequences, the RNA secondary structure is predicted using the NUPACK source code (http://www.nupack.org[21]) and a folding score (FSi) is attributed (between 0 and 1) which measure the similarity between the predicted values and the expected RNA fold (p scores per ASL). Using the {FSi}$_{i = 1…p}$ set, a score is computed for each ASL in the pool by multiplying all corresponding FSi. Subsequently, a new ASL pool is generated for the next generation by only conserving the fittest loops (top scores). The iterative process stops once the number of iterations exceeds a user-defined threshold (20 in this study) and the top score remains unchanged for two consecutive iterations. Between ASLs with similar scores, we favoured those displaying a GC content close to 50% as well as those devoid of consecutive same nucleotide repeats. The iSBHfold algorithm can be accessed at http://apps.molbiol.ox.ac.uk/iSBHfold/cgi-bin/iSBHfold.cgi.

**Data availability.** No data sets were generated during the current study. All data values supporting the experimental conclusions are shown either in main or Supplementary Figures (source data are available from corresponding author). Sequences and predicted RNA secondary structures of all SBH constructs used in this study are reported in Supplementary Fig. 12.

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

## Acknowledgements

We thank Kevin Clark and Paul Sopp (WIMM) for help with the flow cytometry experiments, and Toni A. Baeumler, Mike Barnkob, Andrew Bassett, Diana Chin, Yale S. Michaels, Bruno R. Steinkraus, Markus Toegel and Qianxin Wu for providing critical comments on the manuscript. Q.R.V.F. is supported by a Wellcome Trust PhD studentship. R.L. was a visiting student under Erasmus + program from the University of Wuerzburg. T.A.F. is supported by MRC (G0902418), BBSRC (BB/N006550/1) and Wellcome Trust ISSF (105605/Z/14/Z).

## Author contributions

Q.R.V.F. and T.A.F. conceived the study and designed the experiments; Q.R.V.F. performed the experiments with help from R.L.; Q.R.V.F. analysed results; Q.R.V.F. and T.A.F. wrote the manuscript.

## Additional information

**Competing financial interests:** A patent application (United Kingdom Patent Application No. 1700460.7) related to the SBH system described in this manuscript has been submitted.

**Publisher's note**: 

