## [Peer Review File · Nature Communications]

Reviewers' Comments:

Reviewer #1 (Remarks to the Author)

Ferry et al. "Inducible CRISPR guide RNAs for de novo assembly of transcriptional programs"

Summary

The revised manuscript reports an engineered CRISPR system (CRISPR-TR) that allows inducible transcriptional regulation by controlling guide RNA structure. The original manuscript demonstrated a concept and approach that is likely to be of interest to a broad audience. The authors have satisfactorily addressed most of the major comments. One significant issue remains to be addressed. After the authors address this issue, the manuscript should be published.

Major Comment:

I remain concerned that the "activation score" metric used by the authors to analyze flow cytometry data gives misleadingly large values of "fold-change" between "off" and "on" states. The authors use an activation score where the median fluorescence is multiplied by the % positive cells. As noted by the authors, this value is very sensitive to the choice of threshold for the %positive cutoff, especially when cells are in an "off" state. The method gives nearly zero activation score values when cells are in the off state, and the exact value of this small number depends on the choice of cutoff because so few cells are positive. This means that the fold-change values are unusually large and somewhat arbitrary. Given the arbitrariness of the metric, why is this activation score fold-change value a meaningful number to evaluate?

The observed mean or median fluorescence values from cell populations gated for live, transfected cells still seem to be a more intuitively sensible and meaningful metric for comparison, unless the observed distributions are bimodal with only a small fraction displaying activation.

In the revised manuscript, the authors should be very clear about the significance of the activation score and the fact that the fold-change is arbitrary based on the choice of cutoff. The description in the response to reviewers is reasonably clear, but key details appear to be missing from the revised manuscript. Specifically, this statement from the response is absent in the manuscript:

"To avoid zero values and obtain representative fluorescence measurements in the OFF-state/negative control conditions (where very few cells are activated), the *EXFP*+ve gate is set to allow ~0.1% false positive in the negative control condition (scramble native sgRNA)."

By including and justifying this statement, the authors would have to explicitly address the inherent arbitrariness of the metric, which I think needs to be more clear.

Reviewer #2 (Remarks to the Author)

Dr. Fulga and colleagues have developed an inducible CRISPR transcriptional regulator approach by guide RNA engineering to generate spacer blocking hairpins. Previous concerns mainly concerned the broad applicability of the technology as well as some technical details of this novel application. Dr Fulga and colleagues have now addressed previous concerns of their manuscript in

a resubmission to Nature Communications. I assume that the figures in the response to the reviewer section will be added to the supplement of the min manuscript for the benefit of the reader.

The authors have further emphasized the number of synthetic and endogenous loci. While this is not designed to get insight of the technology at a dozen or so loci, it demonstrates that the loci that were picked work. I still would have preferred to see the performance of the technology at a dozen or so loci but I agree that the current scope is adequate, and meets benchmarks in the field, for publication in Nature Communications.

The authors have done a great job in clarifying the quantitative aspects of their study and have now included more detailed evaluation of their reporter and FACS experiments.

In addition, the added data on cellular toxicity, or lack thereof, is nicely summarized, and ensures the reader of the useful application of the novel inducible system in cellular context.

Overall, I suggest publication of the revised manuscript in Nature Communication.

Reviewers' Comments:

Referee #1:

In the revised manuscript, the authors should be very clear about the significance of the activation score and the fact that the fold-change is arbitrary based on the choice of cutoff. The description in the response to reviewers is reasonably clear, but key details appear to be missing from the revised manuscript.

As requested by the reviewer, we have now converted to the second activation score quantification method. We are also including a detailed description of the activation score calculation and its inherent arbitrary nature in the main manuscript and Methods section.

Referee #2:

I assume that the figures in the response to the reviewer section will be added to the supplement of the main manuscript for the benefit of the reader.

We have now included the data relevant to cellular toxicity in the manuscript as a Supplementary figure 13.